# The Structural Effects of Phosphorylation of Protein Arginine Methyltransferase 5 on Its Binding to Histone H4

**DOI:** 10.3390/ijms231911316

**Published:** 2022-09-26

**Authors:** Rita Börzsei, Bayartsetseg Bayarsaikhan, Balázs Zoltán Zsidó, Beáta Lontay, Csaba Hetényi

**Affiliations:** 1Department of Pharmacology and Pharmacotherapy, Medical School, University of Pécs, 7624 Pécs, Hungary; 2János Szentágothai Research Centre & Centre for Neuroscience, University of Pécs, 7624 Pécs, Hungary; 3Department of Medical Chemistry, Faculty of Medicine, University of Debrecen, 4032 Debrecen, Hungary

**Keywords:** ligand, epigenetics, post-translational modification, signal transduction

## Abstract

The protein arginine methyltransferase 5 (PRMT5) enzyme is responsible for arginine methylation on various proteins, including histone H4. PRMT5 is a promising drug target, playing a role in the pathomechanism of several diseases, especially in the progression of certain types of cancer. It was recently proved that the phosphorylation of PRMT5 on T80 residue increases its methyltransferase activity; furthermore, elevated levels of the enzyme were measured in the case of human hepatocellular carcinoma and other types of tumours. In this study, we constructed the complexes of the unmodified human PRMT5-methylosome protein 50 (MEP50) structure and its T80-phosphorylated variant in complex with the full-length histone H4 peptide. The full-length histone H4 was built in situ into the human PRMT5-MEP50 enzyme using experimental H4 fragments. Extensive molecular dynamic simulations and structure and energy analyses were performed for the complexed and apo protein partners, as well. Our results provided an atomic level explanation for two important experimental findings: (1) the increased methyltransferase activity of the phosphorylated PRMT5 when compared to the unmodified type; (2) the PRMT5 methylates only the free form of histone H4 not bound in the nucleosome. The atomic level complex structure H4-PRMT5-MEP50 will help the design of new inhibitors and in uncovering further structure–function relationships of PRMT enzymes.

## 1. Introduction

Post-translational modification (PTM) is a fundamental mechanism occurring on proteins of different roles in epigenetic regulation [1,2,3,4]. Histone H4 is a building block of the nucleosome, the smallest unit of the chromosome [5]. It also contributes to the epigenetic “histone code” system [6], a combination of PTMs mostly on the N-terminal tail of H4 and other histones. PTMs often involve the covalent attachment of atomic groups to proteins catalysed by different enzymes, also called writers [7].

Protein arginine methyltransferases (PRMT) are writers that add methyl groups to the N-terminal arginine residues of H4 (or other substrates) [8]. It was recently recognised that arginine methylation via PRMTs is associated with several diseases, especially cancer progression [8,9]. Consequently, PRMTs are promising novel drug targets in tumour therapy, as is indicated by the numerous PRMT inhibitors that appeared in preclinical and clinical development [8].

PRMT5 is a member of the PRMT family, catalysing the arginine monomethylation and monomethylation of several non-histone and histone proteins, including histone H2A [10,11,12], H3 [13,14], and H4 [8,11,15,16]. Its activity is linked to mRNA splicing, DNA repair mechanisms, drug resistance, and the regulation of immune cell function [8]. An increased activity and overexpression of PRMT5 was identified in several cancers, making it a promising drug target [8]. PRMT5 is localised in both the cytoplasm and nucleus, in complex with methylosome protein 50 (MEP50), creating an association with numerous partner proteins and several histone and non-histone ligands [15]. However, it was experimentally proved that, likewise to other PRMTs, PRMT5 cannot catalyse the arginine methylation of histones if bound in the nucleosome [11,16].

The phosphorylation and dephosphorylation of tyrosine [17,18] and threonine [19] residues of PRMT5 have an effect on its enzyme activity, and therefore, on the pathomechanism of tumour formation. For example, in the case of hepatocellular carcinoma, the phosphorylation/dephosphorylation of PRMT5 on T80 modulates its methyltransferase activity, and the dephosphorylating myosine phosphatase has a tumour suppressor role [9]. Due to the role of PRMT5 in tumourigenesis, the regulation of its enzymatic activity is the major point of interest. It can be regulated at the molecular level, primarily by the formation of the methylosome complex, containing PRMT5 and its various partners, such as MEP50 [17]. However, the major regulatory action on PRMT5 was described by Rho A activating kinase (ROK) and myosin phosphatase (MP), which also counteract on the T80 phosphorylation site of PRMT5, regulating its methyltransferase activity, both in vitro and in vivo. MP modulates the symmetrical dimethylation of histone core proteins in the cell nucleus via the dephosphorylation of PRMT5 at its activating phosphorylation site, causing changes in gene expression. In tumour cells, the inhibitory phosphorylation of MP is increased, leading to higher phosphorylation levels of PRMT5 at T80 by ROK [9].

The experimental atomic resolution structure of human PRMT5 in complex with MEP50, a methyl-donor ligand and an eight-amino-acid-long histone H4 fragment, was revealed 10 years ago [17]. Some years earlier, structures of non-human PRMT5 were published, as well [20,21]. However, complex structures with the full-length histone H4 have not been published yet.

The aim of this study was to construct the human PRMT5-MEP50 structure in complex with the full-length histone H4 peptide in order to provide a structural explanation for the increased methyltransferase activity of the T80-phosphorylated enzyme (PRMT5_P_), as well as for the inactivity of PRMT5 on nucleosome-bound histone H4.

## 2. Results and Discussion

### 2.1. Unmodified PRMT5 in Complex with the Full-Length H4 Protein

The explanation of the difference in enzymatic activity (activation energy) of the two PRMT5 variants necessitates the atomic resolution structures of the H4-PRMT5 complexes. However, PRMT5(-MEP50) in complex with the full-length histone H4 has not been measured yet (see Appendix A for available PRMT5 structures [17,20,21,22,23,24,25,26,27,28,29,30,31,32,33,34]). Although the number of experimental human PRMT5 complexes increased recently due to its importance in cancer therapy, only one structure (PDB code: 4gqb, [17]) contains an eight-amino-acid-long N-terminal peptide fragment of histone H4 bound to the catalytic domain of PRMT5. The experimental determination of a full-length histone structure may be challenging, partly due to the high flexibility of the N-terminal tail [4]. However, the catalytic domain is positioned far from T80 in space, and therefore, the PRMT5-bound structure of N-terminal tail of histone H4 alone did not provide a sufficient basis for an explanation of the effects of T80 phosphorylation. Thus, the building of the full-length H4 in complex with PRMT5 was necessary to provide an explanation of the effects of T80 phosphorylation. Building the unmodified complex H4-PRMT5(-MEP50) was challenging, as there is no information in the literature about how the full-length H4 fits to PRMT5. The structure of the full-length histone H4 (Figure 1A) is available, e.g., in a nucleosome-bound form (PDB code: 1kx5, [35]). However, the simple superimposition of this full-length nucleosomal H4 to the N-terminal H4 fragment (4gqb) did not result in a collision-free H4-PRMT5 complex. Thus, an in situ, fragment-based construction of the full-length histone H4 structure was performed, using available histone H4 fragment structures, starting from the 8th amino acid of H4 in 4gqb (Methods, Figure 1B). The superimposed H4 fragments were covalently attached, and the H4-PRMT5(-MEP50) complex was energy-minimized and submitted to a 720 ns-long molecular dynamic (MD) simulation.

The schematic energy profile of the human PRMT5-catalysed methylation of H4 is shown in Figure 2A. In the case of the T80 phosphorylation of the enzyme (PRMT5_P_), it can be expected (Introduction) that the activation energy barrier will decrease (Figure 2A) when compared to the unmodified enzyme (PRMT5).

The interaction energy (E_inter_, Methods) between H4 and PRMT5(-MEP50) was calculated for all snapshots of the MD simulation (Figure 2B). Two transient (T1, T2) and two horizontal (plateau) (P1, P2) regions can be distinguished (Figure 2B). E_inter_ showed a relatively quick change up to 120 kcal/mol in the transient regions T1 and T2, and it fluctuated with a maximal amplitude of 50 kcal/mol in the P1 and P2 regions for a longer time period of at least 250 ns (Appendix A). T1 can be assigned as a technical equilibrating region of conformational optimization of the complex, and therefore, it was omitted from the evaluations. At the same time, T2 may correspond to the energy drop connecting the activated and the pre-product states of the enzyme–substrate complex (H4-PRMT5, Figure 2A). The term “pre-product state” is used for a bound H4 conformation appropriately prepared for subsequent methylation, but not yet methylated.

Representative structures of H4-PRMT5 were selected (Method) from both P1 and P2 plateaus and analysed. The interactions between the catalytic domain of PRMT5 and histone H4 N-terminal residues remained stable throughout the MD simulation. At the same time, a considerable change of the H4 structure was observed between the activated (P1) and pre-product (P2) states, also reflected by the aforementioned drop in the total E_inter_ (Figure 2B). In the pre-product structure of H4-PRMT5 (plateau P2), histone H4 interacted with six major regions of PRMT5, including the catalytic sites (e.g., residues E435 and E444), H146, R201, Y304-Q309, D317-Q322, and E483-D491, reflected by the favourable (large negative) E_inter_ contributions of the above regions, listed as bar charts calculated for a representative structure of the P2 complex, and marked with coloured spheres in Figure 3 (Methods). Although Helix 3 of H4 spans over PRMT5:T80, significant interaction could not be measured at T80 (Figure 3). Instead, the C-terminal part of Helix 3 (R67 and D68) showed a remarkable interaction (Figure 3) with the neighbouring PRMT5 residues (marked with blue and magenta in Figure 3). In contrast with the hypothesis of a previous study [16], MEP50 does not play a direct role in histone binding (S4).

### 2.2. The H4-PRMT5-MEP Complex vs. Apo Protein Structures

To examine the conformational changes of histone H4 and PRMT5-MEP50 during complex formation (Section 1), their structures were extracted from the complex and submitted to MD simulations of 1000 and 580 ns, respectively. The root mean square fluctuation (RMSF) of each residue was calculated, and regions with an RMSF higher than 3 Å were collected (Figure 4). Interestingly, two of the four PRMT5 regions (residues 145–148 and 490–491) with an RMSF higher than 3 Å took place in histone H4 binding (Figure 4A). At the same time, residues in these regions had the highest E_inter_ in the pre-product H4-PRMT5 complex structure (Figure 3). Unlike PRMT5, MEP50 showed conformational rigidity (RMSF < 3 Å, Figure 4B), indicating the lack of flexible regions necessary for H4 binding.

In the case of histone H4, the highest RMSF occurred at the linear N-terminal tail (residues 1–33, Figure 4C), including residue R3, methylated by PRMT5. While this region is obviously highly flexible, it is crucial in PRMT5 binding, also indicated by the per-residue E_inter_ contribution (Figure 3). Similarly, the region of residues 39–53 also showed high fluctuation and were involved in the binding of PRMT5 phosphorylated on T80 (PRMT5_P_, see Section 3 for details), but did not show significant interaction with PRMT5 (Figure 3) The third, small region with RMSF > 3 Å was focused on K59, important (Figure 3) in stabilizing the PRMT5 complex structure (Figure 4C).

The calculation of the root mean squared deviation (RMSD) was also performed to estimate the time scale of conformational changes of H4 binding to PRMT5. Considerable changes of the bound structures were measured in terms of Cα RMSD values (Appendix A) of 7.7 (P1) and 9.1 (P2) Å when compared to the representative apo form (last frame). This considerable change in the H4 structure can be attributed to Helix 3, which had a linear conformation in the nucleosome (Figure 5) that is very similar to the representative apo conformation (Figure 5). At the same time, in the pre-product state of the H4-PRMT5 complex (P2), Helix 3 broke in the middle and adopted a V-shaped conformation (Figure 5). This conformational change was further analysed, and the Cα RMSD of Helix H3 was calculated for the unbound H4 MD trajectory, using the pre-product H4 V-shaped conformation (P2) as a reference structure (Appendix A). Helix 3 of unbound H4 showed flexibility centred at H4:G56, resulting in V-shaped conformations (Appendix A) similar to the complexed H4 (Figure 5). The unbound H4 MD simulation showed that a ca. 400 of 1000 ns (Appendix A) time frame is necessary for the conformational change of Helix 3 from the V-shaped to the linear form of the representative apo H4 structure (Figure 5).

### 2.3. Structural Effects of Phosphorylation on T80

The structural explanation of the increased methyltransferase activity of PRMT5_P_ necessitates the building of the atomic resolution structure of the H4-PRMT5_P_-MEP50 complex for a comparison with the unmodified PRMT5 version that is described in Section 1. The H4-PRMT5_P_-MEP50 complex was constructed by adding a phosphate group to residue T80 of the energy-minimized H4-PRMT5 complex structure (Methods). The complex was energy-minimized and subjected to a 720 ns-long MD simulation, and E_inter_ was calculated between H4 and PRMT5_P_-MEP50 along the whole trajectory (Figure 2B).

On average, H4-PRMT5_P_ showed a lower E_inter_ when compared to the unmodified H4-PRMT5, regardless of the method used to select the representative P2 structure (Appendix A). In region P2, the E_inter_ values of both complexes were comparably low (Figure 2B, Appendix A), indicating that both systems reached an energetically favorable (pre-product) state (Figure 2A). The overall lower E_inter_ of H4-PRMT5_P_ for the full trajectory is due to the lack of plateau P1 of a relatively high E_inter_ at H4-PRMT5 (Figure 2B, Appendix A). Thus, in H4-PRMT5_P_, the phosphorylation of T80 resulted in a favourable E_inter_, lowering the activation energy barrier. The lower activation energy also means an increased methyltransferase activity of PRMT5_P_, which was verified experimentally [9].

The abovementioned lowering of E_inter_ comes from the change of interaction network between the phosphorylated T80 (pT80) residue of PRMT5 and histone H4, as reflected by the per-residue E_inter_ analysis (Figure 3). The highest change of E_inter_ occurred on pT80, E320, and K302 of PRMT5_P_ (Figure 3). Residue pT80 formed stable H-bridge interactions with R40 and R45 of histone H4 (Figure 6A, Appendix A). The stabilization of these interactions for several hundreds of nanoseconds is reflected by the corresponding distance plots prepared for the full MD simulation (Appendix A). In the case of unmodified H4-PRMT5, the complex was formed by the C-terminal (R67 and D68) residues of Helix 3 of H4 interacting with a PRMT5 region different from T80 (blue and magenta in Figure 3). These interactions resulted in a V-shaped conformation (Figure 5) of Helix 3 in the H4-PRMT5 complex, while a linear Helix 3 was observed in H4-PRMT5_P_ (Appendix A), similar to the apo form of H4 (Figure 5). This distortion of Helix 3 is an important factor of its unfavourable average E_inter_ (Figure 3, Appendix A) in the case of the unmodified H4-PRMT5 complex. At the same time, binding of Helix 3 in an unchanged, linear form, that is, a “binding competent conformation” [36] contributed to the stronger interaction in H4-PRMT5_P_.

Likewise, to the H4-PRMT5_P_ complex, histone H4 residue R45 is also involved in the interaction of the phosphate groups of nucleosomal DNA. The H4-DNA interactions were listed using an experimental nucleosome structure (PDB code: 1kx5, [35], Methods, Appendix A) and are shown in Figure 6b, with a close-up on the interacting residues marked as sticks. Among the interacting H4 residues (Figure 1A), R45 is one of the most important binding partners of the DNA phosphate groups (the phosphate groups at dT+7 and dG+8 are involved in the interactions with R45, Appendix A, Figure 6b). Such electrostatic interactions are of primary importance in the stabilization of the nucleosome [37]. The abovementioned arginine–phosphate interactions are well-documented for nucleic acid partners [38], due to the ideal geometry, charge distribution, and flexibility of the arginine side-chain [39].

As our model shows that the phosphate group of pT80 residue of PRMT5_P_ interacts with histone H4 residue R45 (Figure 6A), the above interactions (Figure 6B) of R45 with nucleosomal DNA cannot be formed in the presence of PRMT5_P_. This structural conclusion of the present study is in line with the experimental fact that the nucleosome-bound histone H4 is not a substrate of PRMT5-MEP50 [11,16].

## 3. Materials and Methods

### 3.1. In Situ Fragment-Based Construction of H4 in Complex with PRMT5

For building the human H4-PRMT5 complex, the crystal structure of the human PRMT5-MEP50 complex, bound to a histone H4 fragment (1–8 residues), was used as the starting structure (4gqb). The missing amino acids of PRMT5-MEP50 were built by SWISSMODEL online server [40], and terminals were capped. To build the full-length histone H4, peptide fragments from at least the 8th residue were needed. Therefore, all H4 structures linked in the UniProt database [41] under the human H4 UniProt entry (P62805) were checked in the PDB databank [42] and filtered based on the first resolved H4 residue and the length of the experimentally revealed H4 fragment. Structures of the human transcriptional protein (2kwo, [43]) and the nucleosome core particle (3x1v, [44]) bound to histone H4 were chosen, containing H4 residues 2–20 and 16–102, respectively (Figure 2B). After several attempts of superimposing of the abovementioned fragments to the resolved H4-PRMT5 complex, only the backbone alignment (with C, N, O, and Cα atoms) of histone H4 residues, such as K8 and K20, resulted a complex without collision (Figure 2B). The overlapping residues were cut (H4 residues 2–8 and 16–20 from the structures of 2kwo and 3x1v, respectively), and the superimposed H4 fragments were covalently attached in Maestro [45]; the H4-PRMT5(-MEP50) complex was equilibrated by the two-step energy minimization procedure (detailed later).

The root mean square deviation (RMSD) for Cα atoms was calculated during the MD simulation to check the equilibration of the structures. The RMSD of MEP50 fluctuated between 1.5 and 2.0 A (Appendix A), showing a low conformational flexibility; therefore, this structure was used as reference structure in all superposition. The Cα RMSD of PRMT5 and MEP50 was separately calculated after the least square fitting of snapshots to the first MEP50 structure. The RMSD of the unbound H4 structure was also calculated after the least square fitting of snapshots to the assembled H4 structure obtained from the energy-minimized H4-PRMT5 complex during the MD simulation. RMSD was calculated by Equation (1):(1)$$\mathrm{RMSD}\left(t_{1},t_{2}\right)={\left[\frac{1}{M}\sum_{i=1}^{N}m_{i}\|r_{i}\left(t_{1}\right)-r_{i}{\left(t_{2}\right)\|}^{2}\right]}^{\frac{1}{2}}\;;M=\sum_{i=1}^{N}m_{i}$$
where m_i_ is the atomic mass, and r_i_(t) is the position of atom i at time t.

The root mean square fluctuation (RMSF) was also calculated for each residue in the case of the apo PRMT5-MEP50 and unbound H4 structures, after the snapshots were fitted to the same structures like in the RMSD calculation. GROMACS [46] was used for all RMSD (command: *gmx rms*) and RMSF (command: *gmx rmsf*) calculations. The RMSF was calculated by Equation (2):(2)$${\mathrm{RMSF}}_{i}={\left[\frac{1}{T}\overset{T}{\underset{t_{j}=1}{\sum }}\|r_{i}\left(t_{j}\right)-r_{i}^{\mathrm{ref}}\|{}^{2}\right]}^{\frac{1}{2}}$$
where r_i_ is the position of the particle i, T is the time of the MD simulation, and ref denotes the reference position of the particle i.

To build the phosphorylated H4-PRMT5_P_ complex, the phosphate group was covalently attached to T80:PRMT5 by PyMol [47]. The parameters of the phosphorylated threonine were obtained from a previous study [48].

### 3.2. Energy Minimization

Complexes and peptides were submitted to a two-step (steepest descent and conjugate gradient) energy minimization procedure before the MD simulation by GROMACS [46]. Molecules were placed in the centre of a cubic box, with a distance of 10 Å between the box and the solute atoms. The simulation box was filled with TIP3P [49] explicit water molecules and counter ions to neutralize the total charge of the system. The convergence threshold of steepest descent and conjugant gradient step of minimization was set to 100 and 10 kJ mol^−1^ nm^−2^, respectively.

### 3.3. Molecular Dynamic Simulation

The complex, the unbound histone H4, and the apo PRMT5-MEP50 were submitted separately to a 720 ns-, a 1000 ns-, and a 580 ns-long MD simulation, respectively. In all cases, a TIP3P [49] explicit water model with an AMBER99SB-ILDN force field [50] was applied using the GROMACS program package [46], following the two-step energy minimization procedure (described above). Histone H4 and the enzymes could move freely; position restraints were not applied. For temperature-coupling, the velocity rescale and the Parrinello–Rahman algorithm were used. The solute and solvent were coupled separately, with a reference temperature of 310.15 K and a coupling time constant of 0.1 ps. The pressure was coupled by the Parrinello–Rahman algorithm and a coupling time constant of 0.5 ps, compressibility of 4.5 × 10^−5^ bar^−1^**,** and reference pressure of 1 bar. Particle mesh Ewald summation was used for long-range electrostatics. Van der Waals and Coulomb interactions had a cut-off at 11 Å. Periodic boundary conditions were treated after the finish of the calculations. After each trajectory, the periodic boundary effects were handled, the system was centred in the box, and the target molecules in subsequent frames were fit on the top of the first frame. The final trajectory, including all atomic coordinates of all frames, were converted to portable xdr-based xtc binary files.

### 3.4. Interaction Energy Calculations

The sum of Lennard-Jones (LJ) and Coulomb (Cb) intermolecular interaction energies were calculated [51] (3). The Coulomb term was globally calculated with a distance-dependent dielectric function [52] (4) and Amber partial charges [50,53], with per-residues during the simulations, and was represented as intermolecular interaction energy (E_inter_).
(3)$$E_{inter}=E_{LJ}+E_{Coulomb}=\overset{N_{E}\;N_{S}}{\underset{i,j}{\sum }}\left(\frac{A_{ij}}{r_{ij}^{12}}-\frac{B_{ij}}{r_{ij}^{6}}+\frac{q_{i}q_{j}}{4\pi \varepsilon_{0}\varepsilon_{r}r_{ij}}\right)$$
$$A_{ij}=\varepsilon_{ij}R_{ij}^{12}$$
$$B_{ij}=2\varepsilon_{ij}R_{ij}^{6}$$
$$R_{ij}=R_{i}+R_{j}$$
$$\varepsilon_{ij}=\sqrt{\varepsilon_{i}\varepsilon_{j}}$$
where ε_ij_ is the potential well depth at equilibrium between the ith (substrate) and jth (enzyme) atoms; ε_0_ is the permittivity of vacuum; ε_r_ = 1, relative permittivity; R_ij_ is the inter-nuclear distance at equilibrium between ith (substrate) and jth (enzyme) atoms; q is the partial charge of an atom; r_ij_ is the actual distance between the ith (substrate) and jth (enzyme) atoms; N_E_ is the number of enzyme atoms; N_S_ is the number of substrate atoms.
(4)$$\varepsilon_{r\;}=A+\frac{B}{1+ke^{-\lambda Br}}$$
where B = ε_0_ − A, ε_0_ is the dielectric constant of water at 25 °C, and A, λ, and k are constants.

The top ten residues with the lowest E_inter_ values at both the enzyme and the substrate sites for the unmodified and phosphorylated H4-PRMT5 complexes were chosen and merged to prepare the E_inter_ bar chart (Figure 3).

### 3.5. Selection of Representative Structures by Structural Clustering and Interaction Energy Differences

Representative structures were selected using a structure-based clustering from the following four sets of structures: (i) unmodified H4-PRMT5 complex structures from the P1 and (ii) P2 plateaus; (iii) phosphorylated H4-PRMT5 complex structures from the P1 and (iv) P2 plateaus. The clustering procedure contained the following steps: The average atomic coordinates were calculated for all four set of structures using a bash script, which prints the x, y, and z atomic coordinates of all structures in the set into separate text files. The atomic coordinates were structured into a pdb file format and used as average structures. Finally, the RMSD values between the average structure and each complex were calculated by an in-house program, rmsd, and the structure with the lowest RMSD value was selected as the representative structure.

An E_inter_-based selection of representative structures was also performed in the pre-product state (P2), as E_inter_ should have a similar value in the unmodified and phosphorylated structures. Accordingly, representatives for the P2 section were determined by calculating the E_inter_-differences of the unmodified H4-PRMT5 and H4-PRMT5_P_ complexes of the last twenty-five frames. Structures with the lowest E_inter_-difference were chosen as unmodified and phosphorylated H4-PRMT5 representatives for the P2 section (Figure 2B).

### 3.6. Determination of DNA Binding Domain of H4 in Nucleosome

The structure of the nucleosome core particle (1kx5, [35]) contained two full-length histone H4s (chain ID: B and F). Histone H4 residues within 3.5 Å distance from the DNA chains were collected for both of the H4 peptides, using an in-house program. Amino acids taking place in DNA binding in the case of both peptides were determined as DNA binding domains of H4 in the nucleosome (Appendix A).

## 4. Conclusions

A three-dimensional structure of the unmodified and phosphorylated human PRMT5-MEP50, in complex with the full-length histone H4 protein, was modeled. Molecular dynamic simulations and subsequent analyses provided an atomic level explanation for two important experimental findings: (1) the increased methyltransferase activity of the phosphorylated PRMT5 when compared to the unmodified type [9]; (2) the PRMT5 methylates only the free form of histone H4 not bound to the nucleosome [11,16,20]. We expect that our findings will foster the design of new inhibitors and help in uncovering further structure–function relationships of PRMT enzymes.

## Figures and Tables

**Figure 1 ijms-23-11316-f001:**
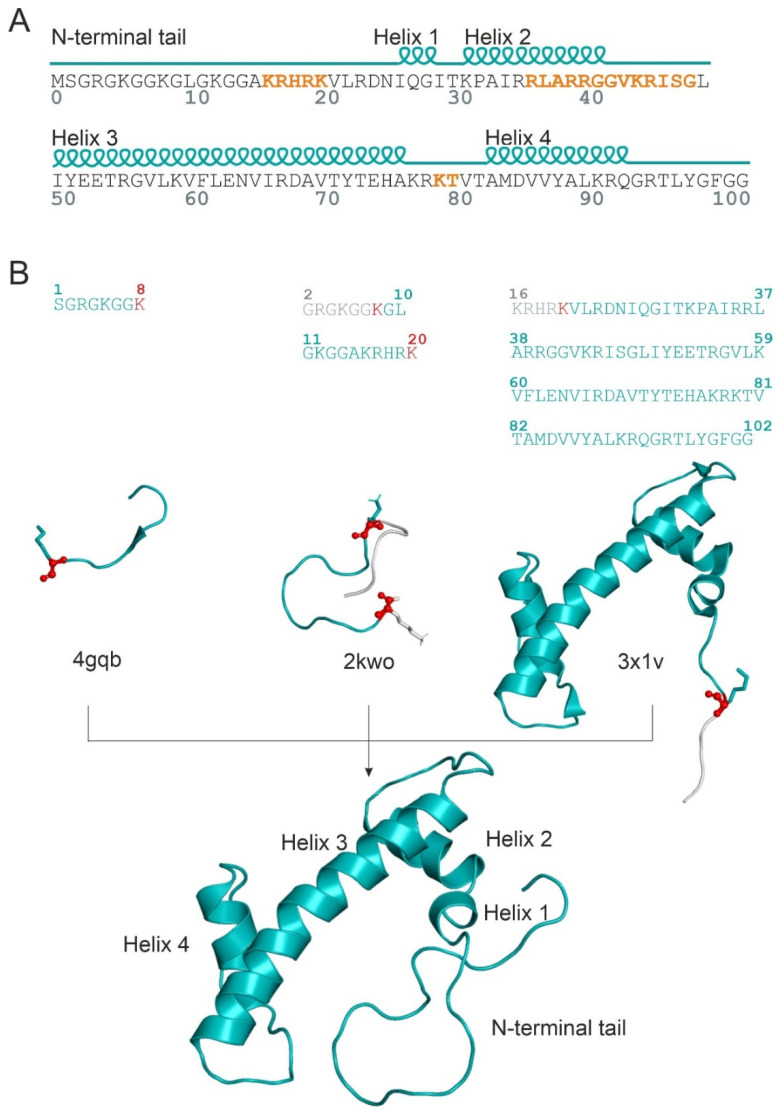
(**A**) Sequence and secondary structure of histone H4 (Uniprot code: P62805). DNA binding regions are highlighted with orange; (**B**) the process of in situ, fragment-based construction of the full-length histone H4 (down, cartoon, teal) by the usage of peptide fragments of different lengths obtained from experimental structures (PDB codes: 4gqb, 2kwo, and 3x1v). Residues (sticks, teal) and backbone atoms (N, Cα, C, O, and spheres) used for alignment are highlighted with red, while the overlapped regions (grey) were cut.

**Figure 2 ijms-23-11316-f002:**
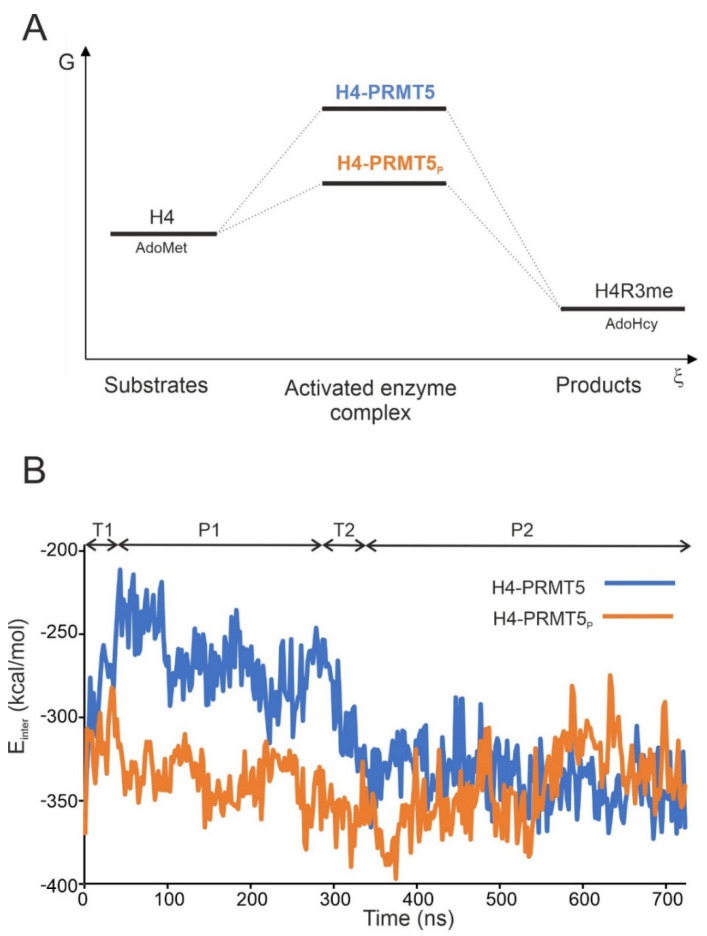
(**A**) Schematic free energy (G) vs. reaction coordinate (ξ) profile of human PRMT5-catalysed methylation of histone H4. Substrate histone H4 peptide and the methyl donor S-adenosyl-L-methionine (AdoMet) are marked. The free energy of the activated enzyme complexes is relatively high. However, the phosphorylated enzyme complex (H4-PRMT5_P_, orange) has to cross a lower energy barrier than the unmodified complex (H4-PRMT5, blue). Products R3-methylated histone H4 (H4R3me) and S-adenosyl-L-homocysteine (AdoHcy) are also shown; (**B**) intermolecular interaction energy (E_inter_) changed during the 720 ns-long MD simulation for the unmodified (H4-PRMT5, blue) and the T80-phosphorylated (H4-PRMT5_P_, orange) enzyme complexes. The plot includes two transient (T1, T2) and two plateau (P1, P2) regions. T1 is an equilibration section for the conformational optimalization, while T2 refers to the energy drop between the activated (P1) and the pre-product (P2) states of the enzyme–substrate complexes.

**Figure 3 ijms-23-11316-f003:**
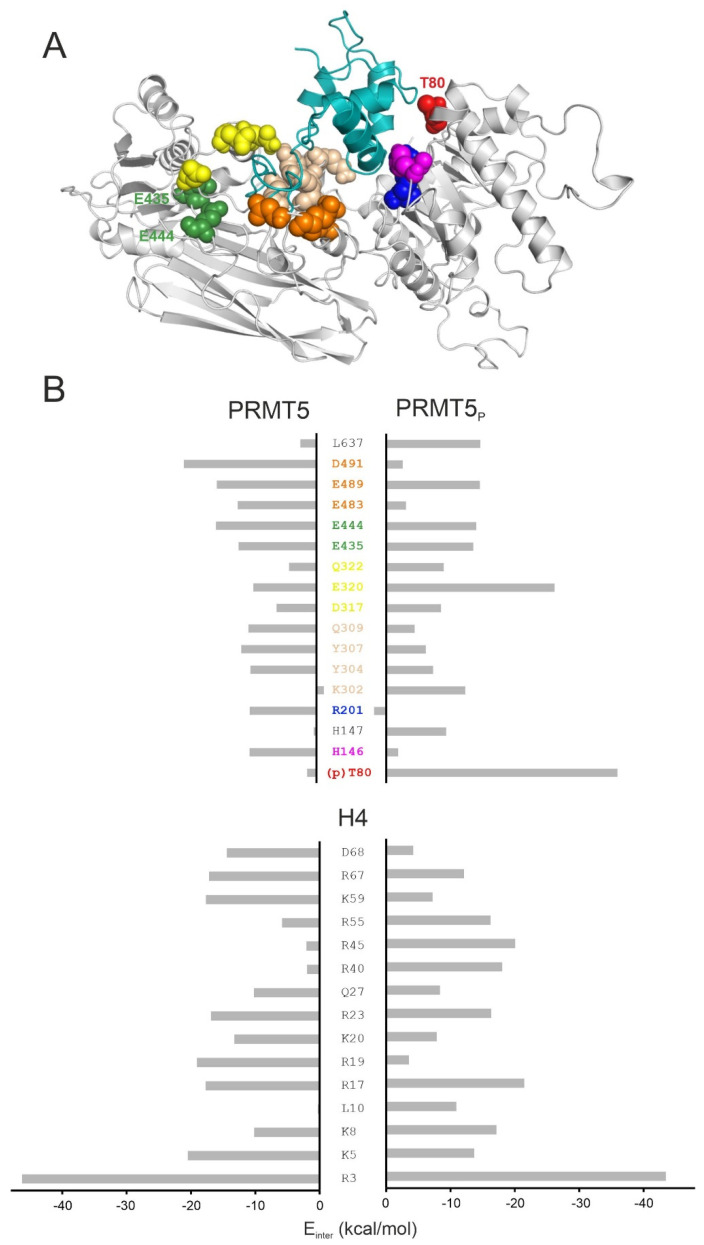
(**A**) The representative structure of the unmodified H4-PRMT5 complex in the pre-product state (P2 in Figure 1B). Note that MEP50 is not shown in this figure due to space restrictions. For the structure of the full H4-PRMT5-MEP50 complex, please refer to Appendix A. Enzyme residues with the lowest E_inter_ are highlighted with colours and spheres, represented in both the structure and the energy bar chart; (**B**) E_inter_ values are calculated for the enzyme and the histone H4 residues in the unmodified PRMT5 and the phosphorylated (PRMT5_P_) complexes in the pre-product (P2) state.

**Figure 4 ijms-23-11316-f004:**
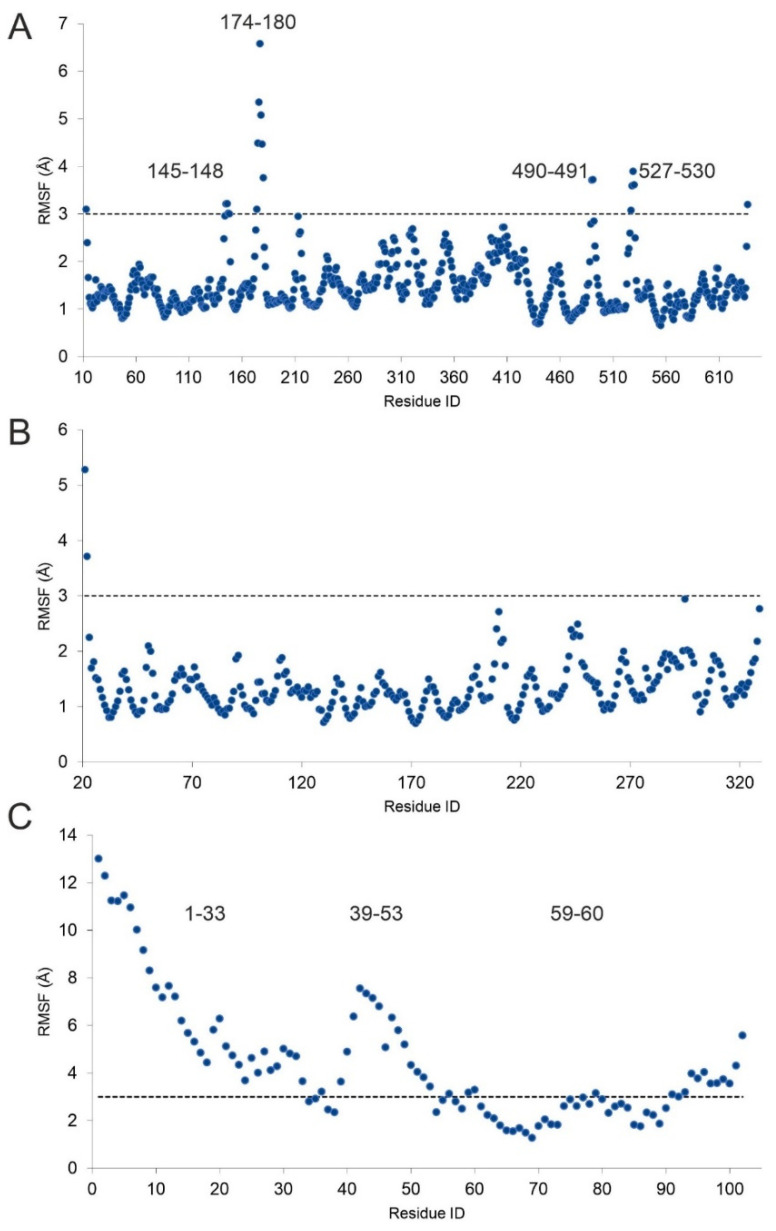
Root mean square fluctuation (RMSF) of each residue in the apo enzyme PRMT5 (**A**), MEP50 (**B**), and the unbound H4 (**C**) during 580, 580, and 1000 ns-long MD simulations, respectively. Regions with higher than 3 Å fluctuation are also marked at the top of the corresponding peaks.

**Figure 5 ijms-23-11316-f005:**
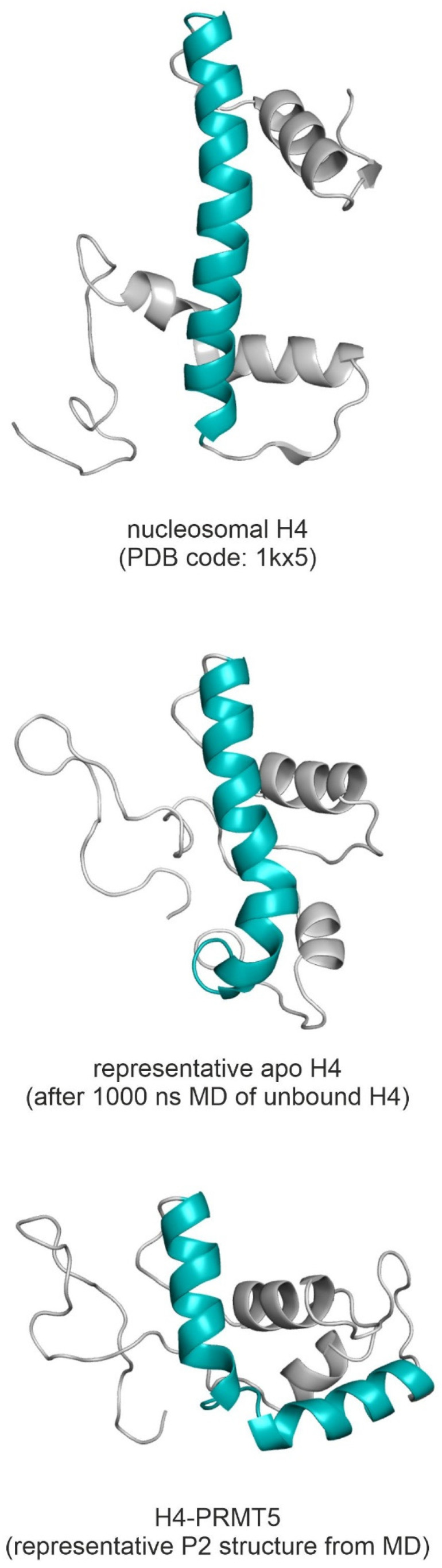
Histone H4 conformations (cartoon, grey) in the nucleosomal, unbound (apo) and PRMT5-complexed forms. Helix 3 (highlighted in teal) adopts a V-shaped conformation in the PRMT5-complexed structure, while it is mostly linear in the nucleosomal and apo forms.

**Figure 6 ijms-23-11316-f006:**
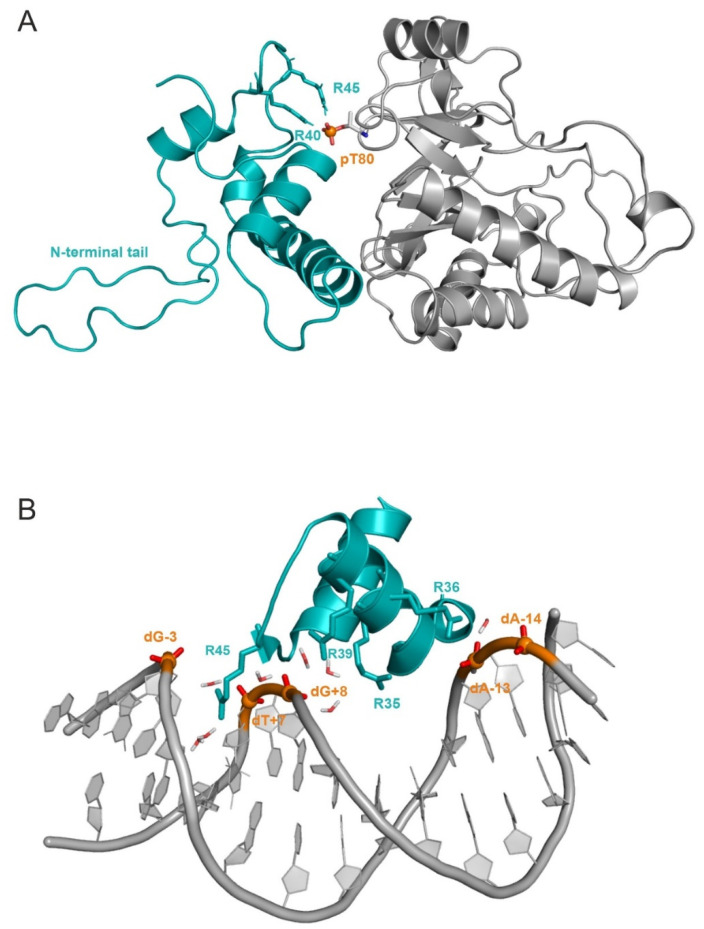
(**A**) Histone H4 (teal, cartoon) bound to PRMT5 (cartoon, grey) phosphorylated on T80 (pT80). Main anchoring residues, R40, R45, and pT80 are represented with sticks. MEP50 and certain parts of PRMT5 and H4 are not shown; (**B**) histone H4 (teal, cartoon) bound to DNA (grey, cartoon) in the experimental nucleosome structure (PDB code: 1kx5). An abridged structure of H4 is shown. Anchoring points are highlighted with sticks. Phosphates (orange) are represented with spheres in (**A**,**B**). In the 1kx5 structure, histone H4 and DNA are depicted as chain ID(s) F and I, and J, respectively. Note that histone H4 is embedded into the nucleosome, except for its N-terminal tail hanging out. The DNA binding domain of H4 is composed by K16-K20, based on the UniProt database (numbering according to PDB), which was identified in the nucleosome complex (1kx5), as well. However, residues before this region (Appendix A) can also create interactions with DNA, due to the flexibility of the N-terminal tail. Furthermore, other H4 regions, such as R36-G48 and K79-T80, were also found to interact with DNA (Figure 1A, Appendix A).

## Data Availability

Not applicable.

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
