# Peer review of "The Structural Effects of Phosphorylation of Protein Arginine Methyltransferase 5 on Its Binding to Histone H4"

_ijms, 2022, doi:10.3390/ijms231911316_

Round 1

Reviewer 1 Report

In the paper "The Structural Effects of Phosphorylation of Protein Arginine Methyltransferase 5 on Its Binding to Histone H4" of Boerzsei et al., the authors provide a structural model of  the human PRMT5-MEP50 structure in complex with the full-length histone H4 peptide to provide the structural explanation for the in-creased methyltransferase activity of T80-phosphorylated enzyme (PRMT5P), and the in-activity of PRMT5 on nucleosome-bound histone H4.

So far no experimental structure was calculated for the complex PRMT5/MEP50 in complex with full length histone H4 and the authors use Molecular Dynamics calculations to fill this gap. The structure of the complex is of general interest because arginine metilation of H4 via PMRT5 is associated with several diseases included cancer and the knowledge of the structure of the complex could help design of new inhibitors, and uncovering further structure-function relationships ofPRMT enzymes.

The paper is clearly written but before being accepted for publication the authors should address  the following points:

The analysis of MD data is based mainly  on RMSD, RMSF and interaction energies, the authors should think of some other kind of analysis like cluster analysis or PCA analysis of the complexes.

For sake of clarity would be better to have before the description of the procedure with which the structural model of the complex was obtained and after the discussion on the MD calculations and interaction energy measurements, consequently figure 1 should be swapped with figure 2.

Author Response

Reviewer 1

In the paper "The Structural Effects of Phosphorylation of Protein Arginine Methyltransferase 5 on Its Binding to Histone H4" of Boerzsei et al., the authors provide a structural model of the human PRMT5-MEP50 structure in complex with the full-length histone H4 peptide to provide the structural explanation for the in-creased methyltransferase activity of T80-phosphorylated enzyme (PRMT5P), and the in-activity of PRMT5 on nucleosome-bound histone H4.

So far no experimental structure was calculated for the complex PRMT5/MEP50 in complex with full length histone H4 and the authors use Molecular Dynamics calculations to fill this gap. The structure of the complex is of general interest because arginine methylation of H4 via PMRT5 is associated with several diseases included cancer and the knowledge of the structure of the complex could help design of new inhibitors and uncovering further structure-function relationships of PRMT enzymes.

The paper is clearly written but before being accepted for publication the authors should address the following points:

Comment 1

The analysis of MD data is based mainly on RMSD, RMSF and interaction energies, the authors should think of some other kind of analysis like cluster analysis or PCA analysis of the complexes.

 Response 1

The representative structures regarding Plateau 1 (P1) and Plateau 2 (P2) regions of intermolecular energy change – time plot (Fig. 1) were actually selected by a cluster analysis based on structural similarity. However, it has not been fully detailed and emphasised in Section Methods. In agreement with the suggestion of the Reviewer, the Supplementary Material and Section 3.5 of Methods were completed with additional details of the results and the detailed description of the analysis, respectively.

Comment 2

For sake of clarity would be better to have before the description of the procedure with which the structural model of the complex was obtained and after the discussion on the MD calculations and interaction energy measurements, consequently figure 1 should be swapped with figure 2.

 Response 2

Indeed, swapping of the two figures increases the clarity of the discussion. The changes were introduced in the main manuscript according to the suggestion of the Reviewer.

 We thank Reviewer 1 for careful evaluation of our manuscript and his/her helpful specific comments.

Reviewer 2 Report

The manuscript ‘The structural effects of phosphorylation of protein arginine methyltransferase 5 on its binding to histone H4’ by Börzsei et al, describes 3D modeling of Histone H4 and H4-PRMT5 complex in phosphorylated and unphosphorylated form. Methyltransferase activity and binding preference to H4 are discussed based on MD simulation results. The manuscript is written very well. The investigation strategy is well-designed and pursued. However, a few minor comments I have as below:

1. Page5, Line121: backbone atoms can be written in well-known order as (N, Ca, C, O)

2. Line123:H4 can be replaced with H4.

3. Line264: Uniprot database is known as sequence database. That includes structural database links, e.g. RCSB PDB. Rephrase the sentence.

4. Line 364: It will be better if you replace “The atomic resolution structure….. was constructed’ with Three-dimensional (or 3D) structure…. was modeled.

Author Response

Reviewer 2

 The manuscript ‘The structural effects of phosphorylation of protein arginine methyltransferase 5 on its binding to histone H4’ by Börzsei et al, describes 3D modeling of Histone H4 and H4-PRMT5 complex in phosphorylated and unphosphorylated form. Methyltransferase activity and binding preference to H4 are discussed based on MD simulation results. The manuscript is written very well. The investigation strategy is well-designed and pursued. However, a few minor comments I have as below:

Comment 1

Page5, Line121: backbone atoms can be written in well-known order as (N, Ca, C, O)

Response 1

According to the suggestion of the Reviewer 2, the order of backbone atoms was corrected (line 121) to N, Ca, C, O.  

 Comment 2

Line123:H4 can be replaced with H4.

Response 2

Thank you for the observation, the subscript of H4 was corrected.

Comment 3

Line264: Uniprot database is known as sequence database. That includes structural database links, e.g. RCSB PDB. Rephrase the sentence.

Response 3

In agreement with Reviewer 2 we rephrased that sentence. Therefore, all H4 structures linked in the UniProt Database [28] under the human H4 UniProt Entry (P62805) were checked in the PDB DataBank [29] and filtered based on the first resolved H4 residue and the length of the experimentally revealed H4 fragment.

Comment 4

Line 364: It will be better if you replace “The atomic resolution structure….. was constructed’ with Three-dimensional (or 3D) structure…. was modeled.

Response 4

The phrases “The atomic resolution” and “constructed” were replaced to “Three-dimensional” and “modelled”, respectively, as suggested by Reviewer 2.

We thank Reviewer 2 for the careful evaluation of our manuscript and his/her helpful specific comments.